# Kawasaki Disease-like Vasculitis Facilitates Atherosclerosis, and Statin Shows a Significant Antiatherosclerosis and Anti-Inflammatory Effect in a Kawasaki Disease Model Mouse

**DOI:** 10.3390/biomedicines10081794

**Published:** 2022-07-26

**Authors:** Yusuke Motoji, Ryuji Fukazawa, Ryosuke Matsui, Noriko Nagi-Miura, Yasuo Miyagi, Yasuhiko Itoh, Yosuke Ishii

**Affiliations:** 1Department of Cardiovascular Surgery, Graduate School of Medicine, Nippon Medical School, Tokyo 113-8603, Japan; yusuke-motoji@nms.ac.jp (Y.M.); show@nms.ac.jp (Y.M.); yosuke-i@nms.ac.jp (Y.I.); 2Department of Pediatrics, Graduate School of Medicine, Nippon Medical School, Tokyo 113-8603, Japan; r-matsui@nms.ac.jp (R.M.); yasuhiko@nms.ac.jp (Y.I.); 3Laboratory for Immunopharmacology of Microbial Products, Tokyo University of Pharmacy and Life Sciences, Hachioji 192-0392, Japan; miuranno@toyaku.ac.jp

**Keywords:** Kawasaki disease, vasculitis, atherosclerosis, statin

## Abstract

Kawasaki disease (KD) is an acute form of systemic vasculitis that may promote atherosclerosis in adulthood. This study examined the relationships between KD, atherosclerosis, and the long-term effects of HMG-CoA inhibitors (statins). Candida albicans water-soluble fraction (CAWS) was injected intraperitoneally into 5-week-old male apolipoprotein-E-deficient (Apo E-/-) mice to create KD-like vasculitis. Mice were divided into 4 groups: the control, CAWS, CAWS+statin, and late-statin groups. They were sacrificed at 6 or 10 weeks after injection. Statin was started after CAWS injection in all groups except the late-statin group, which was administered statin internally 6 weeks after injection. Lipid plaque lesions on the aorta were evaluated with Oil Red O. The aortic root and abdominal aorta were evaluated with hematoxylin and eosin staining and immunostaining. CAWS vasculitis significantly enhanced aortic atherosclerosis and inflammatory cell invasion into the aortic root and abdominal aorta. Statins significantly inhibited atherosclerosis and inflammatory cell invasion, including macrophages. CAWS vasculitis, a KD-like vasculitis, promoted atherosclerosis in Apo E-/- mice. The long-term oral administration of statin significantly suppressed not only atherosclerosis but also inflammatory cell infiltration. Therefore, statin treatment may be used for the secondary prevention of cardiovascular events during the chronic phase of KD.

## 1. Introduction

Kawasaki disease (KD) is an acute inflammatory syndrome predominantly affecting infants, causing systemic vasculitis [1]. In Japan, the total number of patients with KD has exceeded 360,000 thus far. Of these patients, 150,000 were over 20 years of age, and 15,000 were over 40 years (from 1964 to 31 December 2016) [2,3]. Among adult patients with KD, it has been reported that vasculitis might induce the early progression of atherosclerosis [4,5,6]. The pathological image of chronic KD cardiovascular sequelae is mainly composed of sclerotic lesions after vasculitis, which is pathologically different from usual atherosclerotic lesions [6,7]. However, it has been demonstrated that chronic vascular endothelial cell damage persists in patients with KD due to the tissue infiltration of inflammatory cells such as macrophage and activation of the immune system by releasing inflammatory cytokines [5,6,8]. Endothelial dysfunction could be said to share a common ground with ordinary atherosclerosis [5,6,8,9].

We have reported findings of prolonged vascular endothelial dysfunction and vascular aging in a pediatric specimen of a human KD coronary artery aneurysm. These findings were similar to early-stage features in adult atherosclerotic lesions [10]. Arditi et al., using an animal model of KD-like vasculitis caused by Lactobacillus casei cell wall bacterial components (LCWE), also reported that complications of dyslipidemia, a risk factor of atherosclerosis, and KD could increase the risk of early-onset atherosclerosis [11]. Currently, several small case studies have reported that HMG-CoA inhibitors (statins) have anti-inflammatory effects and improve vascular endothelial cell function [12,13]. Consequently, Japanese and U.S. treatment guidelines for the remote stage of patients with KD and coronary artery aneurysms recommend the clinical use of statins to prevent atherosclerosis and the restorative effect of coronary artery damage [2,14]. However, there is little evidence for oral statin administration as the basic and clinical mechanisms of the effects of statin have not been elucidated. We have studied the inflammatory response and management of acute KD using the Candida albicans water-soluble fraction (CAWS) model of KD-like vasculitis [15]. CAWS is a substance similar to Candida albicans extracted from the feces of patients with KD. CAWS can be injected into animal models to cause coronary arteritis, vasculitis, and myocarditis similar to KD [16,17]. This study aimed to examine whether a pre-existing KD vasculitis accelerated the development of atherosclerosis in young adulthood and how statins might have therapeutic effects on the prevention and development of atherosclerosis. We created a KD-like vasculitis model of atherosclerosis with CAWS in apolipoprotein-E-deficient (Apo E-/-) mice, an animal model of atherosclerosis, and evaluated whether CAWS vasculitis was a factor promoting atherosclerosis development in the remote stage. We also investigated the therapeutic effect of statins on atherosclerosis caused by KD-like vasculitis.

## 2. Materials and Methods

The study was performed in accordance with the Guide for the Care and Use of Laboratory Animals published by the US National Institutes of Health (NIH publication number 85-23, revised 1996). The study protocol was approved by the Animal Care and Use Committee of Nippon Medical School (approval number: 2020-029).

**1)** 
**Animals**


Five-week-old male C.KOR/StmSlc-ApoE^shl^ mice were purchased from Sankyo Labo Service Co., Ltd. (Tokyo, Japan). All mice were maintained under specific pathogen-free conditions according to the guidelines for animal care of the National Institute of Infectious Diseases in Tokyo [18]. To prevent any gender effects, we used only male mice. Water and food were available ad libitum. Mice were fed a normal diet until 7 weeks of age and were exchanged to a high-fat diet containing 0.15% cholesterol after 7 weeks of age.

**2)** 
**Preparation of CAWS and Statin**


CAWS was prepared from the C. albicans strain NBRC1385 according to a previously reported method [19]. Briefly, 5 L of C-limiting medium was maintained in a glass incubator for two days at 27 °C, while air was supplied at a rate of 5 L/min, and the mixture was swirled at 400 rpm. Following culture, an equal volume of ethanol was added. After standing the mixture overnight, the precipitate was collected and dissolved in 250 mL of distilled water. Ethanol was also added. Subsequently, the mixture was allowed to stand overnight again. The precipitate obtained was collected and dried with acetone to obtain CAWS.

The HMG-CoA inhibitor, Atorvastatin Calcium Hydrate (atorvastatin), was provided by Sankyo Ltd. (Tokyo, Japan). Atorvastatin was crushed and dissolved in 0.5 *w*/*v*% Methylcellulose 400 Solution, Sterilized (WACO FUJIFILM CORPORATION, Tokyo, Japan). Statin was started orally at the same time as the high-fat diet. The latter was at a dose of 10 mg/kg/day from 7 weeks of age. The oral dose for mice was determined according to the clinical dose in humans. To eliminate errors in statin feeding due to individual taking differences, statins were administered directly orally daily using an oral sonde.

**3)** 
**Experimental Procedures**


Mice were divided into the following four groups for comparison (Figure 1). Mice were sacrificed at 6 (11 weeks of age) and 10 (15 weeks of age) weeks after CAWS injections. The life cycle of mice is assumed to be much faster than that of humans, and male mice generally reach sexual maturity at 8 weeks of age. In terms of human age, 5 weeks of age corresponds to childhood, 7 weeks to 10–15 years of age, 11 weeks to adolescence to middle age, and 15 weeks to middle age to early adulthood.

(1) Control group: Five-week-old Apo E-/- mice were injected intraperitoneally with phosphate-buffered saline (PBS) instead of CAWS. Instead of statins, the same volume of methylcellulose solution was orally administered to mice daily in the same manner.

(2) CAWS group: CAWS (4 mg/mouse) was injected intraperitoneally into 5-week-old Apo E-/- mice for 5 consecutive days, as done in the study of Hashimoto et al. [15]. Instead of statin, mice were orally injected with a methylcellulose solution daily in the same manner.

(3) CAWS+statin group: Mice were orally administered statin daily from 2 weeks after CAWS administration (from 7 weeks of age) until the last day of the experiment.

(4) Late-statin group: Mice were started on statin orally from 6 (11 weeks of age) weeks after CAWS administration until the last day of the experiment. Since statins have no indication for infancy in clinical practice, we designed a model in which statins are administered from young adulthood.

**4)** 
**Assessment of atherosclerotic lesions in the aorta**


Mice were anesthetized, and the aortas were excised from the aortic arch to the iliac bifurcation. Whole aortas en face were prepared and stained with Oil red O. Lesion areas were quantified and analyzed using the hybrid cell count system (KEYENCE) and KEYENCE BZX analyzer (Osaka, Japan). Image analysis was performed by a trained observer blinded to treatment [20]. The lipid-stained plaque area in the aorta en face preparations was expressed as a percent of the aortic surface area [11].

**5)** 
**Assessment of the area of aortic root horizontal transection**


Aortic roots were embedded in paraffin and identified in serial sections (5 μm), and stained with hematoxylin-eosin (HE) as described in our early publication [15]. The abdominal aorta below the renal artery was harvested and embedded in paraffin. Serial sections (5 µm) were prepared similarly and stained with HE. The area (mm^2^) of inflammatory cell infiltration was measured, and the ratio of the inflammatory area to the total tissue area on the aortic root was calculated.

**6)** 
**Evaluation of macrophage cell and TGFβ receptor expression in the aortic root**


Sections of the aortic root were analyzed immunohistochemically for the presence of macrophage cells and transforming growth factor (TGF) β receptor expression. Immunostaining was performed to identify macrophage cells and their fractions, which represent inflammatory cells. Immunostaining for TGFβ receptors was also performed because it has been reported that TGFβ signaling is involved in vascular remodeling in KD vasculitis [21]. Accordingly, the following sheep anti-rabbit antibodies were used: anti-Galectin 3 (MAC-2) antibody (1/250, 60 min RT, Abcam; ab76245, UK), a specific marker for macrophages; anti-CD80 (Abcam; ab215166, UK), a marker specific for M1; anti-Mannose receptor (CD206) antibody (1/10,000, 30 min RT, Abcam; ab64693, UK), a marker specific for M2; and anti-TGF**β** receptor II antibody (1/500, 60 min RT, Abcam; ab186838, UK). The sections were further treated with the secondary antibodies and developed using HRP-conjugated DAB substrate (Abcam; ab236446, UK).

**7)** 
**Evaluation of macrophage cells in the abdominal aorta**


For abdominal aorta specimens, an anti-galectin 3 (MAC-2) antibody (1/250, 60 min RT, Abcam; ab76245, UK) was used to stain macrophages. Similarly, HRP-conjugated DAB substrate (Abcam; ab236446, UK) was used for staining.

**8)** 
**Serological Evaluation**


Serum samples collected were stored at −20 Celsius degrees until analysis.

Sandwich enzyme-linked immunosorbent assays were used to detect the plasma levels of high-sensitivity C-reactive protein (hs-CRP) (MyBioSource, San Diego, CA, USA; hs-CRP elisa kit, MBS262829) and a low-density lipoprotein/very low-density lipoproteins (LDL/VLDL) (Abcam, San Francisco, CA, USA; cholesterol Assay Kit-HDL and LDL/VLDL, ab65390) according to the manufacturers’ instructions.

**9)** 
**Statistical Analysis**


Statistical data were expressed as median (upper and lower quartiles) or mean ± standard deviation. Statistical analyses were performed using JMP statistical software version 16 (SAS Institute Inc., Cary, NC, USA). The Kruskal-Wallis test was used to analyze statistical differences among groups. When significance was detected, the Wilcoxon test was used as a post-hoc test to compare values between both groups. A *p*-value < 0.05 was considered statistically significant. In all correlation analyses, Spearman’s rank correlation was used, and Spearman’s coefficients are denoted by ρ.

## 3. Results

**1)** 
**Oil red O staining of the entire aorta (Figure 2)**


All mice treated with CAWS administration had accelerated aortic plaque lesion formation. While the plaque in the ascending aorta and descending thoracic aorta was relatively mild, extensive plaque was observed in the abdominal aorta and below.

The aortic plaque coverage ratio was significantly higher in the CAWS group than in the control group at both 6 and 10 weeks after CAWS administration (6 weeks: 5.2 ± 3.2% vs. 20.6 ± 5.9% [*p* = 0.037]; 10 weeks: 6.5 ± 2.4% vs. 33.0 ± 7.6% [*p* = 0.012]; Table 1 and Figure 2a,b).

Compared to the CAWS group, the CAWS + statin group had a decreasing trend after 6 weeks of CAWS administration and a significant decrease 10 weeks after CAWS administration (6 weeks: 20.6 ± 5.9% vs. 12.7 ± 6.1% [*p* = 0.111]; 10 weeks: 33.0 ± 7.6% vs. 17.5 ± 3.9% [*p* = 0.012]; Table 1 and Figure 2a,b).

Furthermore, the aortic plaque coverage ratio in the late-statin group (21.0 ± 4.0%) was significantly lower than that in the CAWS group at 10 (*p* = 0.022) but equivalent to 6 weeks (*p* = 0.676).

**2)** 
**Evaluation**
**of**
**Aortic root samples**


**2a)** 
**Inflammatory cells were evaluated by HE staining (Figure 3)**


HE staining showed severe inflammatory cell invasion at the aortic root and pericoronary arteries in all CAWS-administered individuals. The inflammatory cell invasion area ratio was significantly higher in the CAWS group than in the control group 6 weeks after CAWS administration and remained unchanged 10 weeks after CAWS (6 weeks: 2.7 ± 1.0% vs. 15.5 ± 1.2% [*p* = 0.030]; 10 weeks: 2.0 ± 0.5% vs. 16.9 ± 1.3% [*p* = 0.052]; Table 1 and Figure 3a,b). Compared to the CAWS group, the CAWS+statin group showed a significant decrease at both 6 and 10 weeks after CAWS (6 weeks: 15.5 ± 1.2% vs. 4.8 ± 1.2% [*p* <0.001]; 10 weeks: 16.9 ± 1.3% vs. 6.0 ± 2.0% [*p* = 0.030]; Table 1 and Figure 3a,b). The inflammatory cell invasion area ratio in the late-statin group significantly decreased to 6.9 ± 1.1% compared to the CAWS group (*p* = 0.030) but did not show a significant difference compared to values in the CAWS+statin group (*p* = 0.471).

**2b)** 
**Immunostaining of macrophage cells with anti-Galectin 3 (MAC-2) antibody (Figure 3)**


A large number of macrophage cells remained at the aortic root and pericoronary arteries of all individuals treated with CAWS, even after 10 weeks of CAWS administration. The macrophage cell invasion area ratio significantly increased in the CAWS group compared to the control group at 6 weeks after CAWS administration and slightly decreased at 10 weeks after CAWS administration (6 weeks: 0.3 ± 0.1% vs. 10.2 ± 2.2% [*p* = 0.030]; 10 weeks: 0. 7 ± 0.4% vs. 6.5 ± 1.1% [*p* = 0.052]; Table 1 and Figure 3a,b).

The macrophage cell invasion area ratio of the CAWS+statin group significantly decreased at both 6 and 10 weeks after CAWS compared to values in the CAWS group (6 weeks: 10.2 ± 2.2% vs. 3.5 ± 0.4% [*p* = 0.030]; 10 weeks: 6.5 ± 1.1% vs. 2.5 ± 0.7% [*p* = 0.030]; Table 1 and Figure 3a,b).

The late-statin group had a macrophage cell invasion area ratio of 3.0 ± 0.8%, significantly lower than that of the CAWS group (*p* = 0.030). There was no significant difference between the CAWS+statin group (*p* = 0.471).

**2c)** 
**Immunostaining of macrophage M1 cells with anti-CD80 antibody**


The macrophage M1 cell invasion area ratio significantly increased in the CAWS group compared to the control group at both 6 and 10 weeks after CAWS administration (6 weeks: 0.6 ± 0.3% vs. 11.4 ± 2.5% [*p* = 0.030]; 10 weeks: 1.0 ± 0.3% vs. 6.3 ± 1.3% [*p* = 0.030]; Table 1). The macrophage M1 cell invasion area ratio was significantly lower in the CAWS+statin group than in the CAWS group at both 6 and 10 weeks (6 weeks: 11.4 ± 2.5% vs. 1.9 ± 0.4% [*p* = 0.030]; 10 weeks: 6.3 ± 1.3% vs. 1.9 ± 0.3% [*p* = 0.030]; Table 1). The macrophage M1 cell invasion area ratio for the late-statin group was 2.1 ± 0.7%, which was significantly lower than that in the CAWS group (*p* = 0.030) but not statistically different compared to values in the CAWS + statin group (*p* = 0.883).

**2d)** 
**Immunostaining of macrophage M2 cells with the anti-CD206 antibody**


The macrophage M2 cell invasion area ratio was significantly higher in the CAWS group than in the control group at both 6 and 10 weeks after CAWS administration (6 weeks: 0.2 ± 0.2% vs. 1.1 ± 0.5% [*p* = 0.030]; 10 weeks: 0.3 ± 0.1% vs. 1.0 ± 0.4% [*p* = 0.052]; Table 1).

The macrophage M2 cell invasion area ratio was significantly higher in the CAWS+statin group than in the CAWS group 6 weeks after CAWS administration. This value peaked at 10 weeks after CAWS administration (6 weeks: 1.1 ± 0.5% vs. 2.1 ± 0.3% [*p* = 0.030]; 10 weeks: 1.0 ± 0.4% vs. 1.8 ± 0.3% [*p* = 0.066]; Table 1).

The macrophage M2 cell invasion area ratio in the late-statin group was 0.9 ± 0.2%, which was not significantly different from either the CAWS or CAWS+statin group (*p* = 0.885 and *p* = 0.066, respectively).

**2e)** 
**M2/M1 macrophage cell ratio**


The M2/M1 macrophage cell ratio in the same sample was calculated; compared to the control group, the CAWS group showed a significant decrease at 6 weeks after CAWS administration and no significant difference at 10 weeks (6 weeks: 0.4 ± 0.1 vs. 0.1 ± 0.1 [*p* = 0.030]; 10 weeks: 0.3 ± 0.1 vs. 0.2 ± 0.1 [*p* = 0.112]; Table 1). The M2/M1 macrophage cell ratio was significantly higher in the CAWS+statin group than in the CAWS group both at 6 and 10 weeks after CAWS administration (6 weeks: 0.1 ± 0.1 vs. 1.1 ± 0.3 [*p* = 0.030]; 10 weeks: 0.2 ± 0.1 vs. 1.0 ± 0.2 [*p* = 0.030]; Table 1). The M2/M1 ratio for the late-statin group was 0.5 ± 0.2, which was significantly higher than that in the CAWS (*p* = 0.030) and significantly lower than that of the CAWS+statin group (*p* = 0.030).

**2f)** 
**Immunostaining with anti-TGFβ receptor II antibody**


The TGF**β** receptor II expression area ratio was significantly higher in the CAWS group than in the control group both after 6 and 10 weeks of CAWS administration (6 weeks: 1.1 ± 0.7% vs. 25.7 ± 3.9% [*p* = 0.030]; 10 weeks: 0.4 ± 0.3% vs. 17.5 ± 2.5% [*p* = 0.030]; Table 1). The CAWS+statin group showed a significant decrease in both 6 and 10 weeks after CAWS administration compared to the CAWS group (6 weeks: 25.7 ± 3.9% vs. 5.7 ± 1.4% [*p* = 0.030]; 10 weeks: 17.5 ± 2.5% vs. 6.2 ± 1.8% [*p* = 0.020]; Table 1). The TGF**β** receptor II expression area ratio in the late-statin group was 7.6 ± 1.4%, which was significantly lower than that in the CAWS group (*p* = 0.030), but not significantly different from that in the CAWS+statin group (*p* = 0.540).

**3)** 
**Evaluation of Infrarenal abdominal aorta samples**


**3a)** 
**Inflammatory cells evaluated by HE staining**


The inflammatory cell invasion area ratio was significantly increased in the CAWS group than in the control group both at 6 and 10 weeks after CAWS administration (6 weeks: 3.3 ± 1.2% vs. 9.5 ± 1.0% [*p* = 0.030]; 10 weeks: 3.3 ± 0.7% vs. 11.1 ± 1.9% [*p* = 0.030]; Table 1). The inflammatory cell invasion area was significantly reduced in the CAWS+statin group than in the CAWS group at both 6 and 10 weeks (6 weeks: 9.5 ± 0.9% vs. 4.2 ± 0.4% [*p* = 0.030]; 10 weeks: 11.1 ± 1.9% vs. 4.6 ± 0.9% [*p* = 0.030]; Table 1). The inflammatory cell invasion area ratio in the late-statin group was 5.8 ± 0.6% compared to the CAWS group, showing a decreasing trend (*p* = 0.052).

**3b)** 
**Immunostaining of macrophage cells with anti-galectin 3 (MAC-2) antibody**


Macrophage cells were detected in aneurysm walls of specimens that formed abdominal aortic aneurysms. The macrophage cell invasion area ratio was significantly higher in the CAWS group than in the control group at both 6 and 10 weeks after CAWS administration (6 weeks: 0.2 ± 0.1% vs. 2.0 ± 1.6% [*p* = 0.030]; 10 weeks: 0.2 ± 0.2% vs. 6.3 ± 3.4% [*p* = 0.030]; Table 1). The macrophage cell invasion area ratio was reduced in the CAWS+statin group than in the CAWS group at both 6 and 10 weeks (6 weeks: 2.0 ± 1.6% vs. 1.1 ± 0.2% [*p* = 1.000]; 10 weeks: 6.3 ± 3.4% vs. 2.1 ± 1.9% [*p* = 0.112]; Table 1). The macrophage cell invasion area ratio of the late-statin group was 2.0 ± 2.3%, showing a decreasing trend compared to the CAWS group (*p* = 0.112) and no significant difference compared to the values in the CAWS+statin group (*p* = 1.000).

**4)** 
**S**
**erological examination**


**4a)** 
**Serum LDL/VLDL cholesterol levels in each group**


LDL/VLDL cholesterol levels were significantly higher in the CAWS group than in the control group only at 6 weeks after CAWS administration, yet results were similar between groups at 10 weeks of administration (6 weeks: 472.6 ± 45.0 mg/dL vs. 543.0 ± 6.8 mg/dL [*p* = 0.005]; 10 weeks: 513.3 ± 49.8 mg/dL vs. 532.9 ± 13.9 mg/dL [*p* = 0.379]; Table 1). LDL/VLDL cholesterol levels were significantly lower in the CAWS+statin group than in the CAWS group only 6 weeks after CAWS administration (6 weeks: 543.0 ± 6.8 mg/dL vs. 528.3 ± 11.6 mg/dL [*p* = 0.031]; 10 weeks: 532.9 ± 13.9 mg/dL vs. 532.0 ± 8.6 mg/dL [*p* = 0.810]; Table 1). LDL/VLDL was significantly lower in the late-statin group than in the control group at 10 weeks after CAWS administration (*p* = 0.005).

**4b)** 
**Serum hs-CRP levels in each group**


hs-CRP was significantly higher in the CAWS group than in the control group both at 6 weeks and 10 weeks after CAWS administration (6 weeks: 102.7 ± 20.9 pg/mL vs. 307.4 ± 121.5 pg/mL [*p* <0.001]; 10 weeks: 79.4 ± 17.8 pg/mL vs. 396.3 ± 197.5 pg/mL [*p* < 0.001]; Table 1). hs-CRP was not significantly different between the CAWS and CAWS+statin groups at either 6 or 10 weeks after CAWS administration (6 weeks: 307.4 ± 121.5 pg/mL vs. 248.2 ± 121.5 pg/mL [*p* = 0.318]; 10 weeks: 396.3 ± 197.5 pg/mL vs. 314.6 ± 108.9 pg/mL [*p* = 0.471]; Table 1).

**4c)** 
**Correlations between aortic plaque area, inflammatory cell invasion area, and macrophage invasion area**


Correlations were examined between aortic plaque coverage ratio, inflammatory cell area ratio at the aortic root, and macrophage cell area ratio. A significant correlation was found between the aortic plaque coverage ratio and aortic root inflammatory cell invasion area ratio (ρ = 0.786; *p* <0.0001) and between the aortic plaque coverage ratio and aortic root macrophage area ratio (ρ = 0.641; *p* = 0.0004) (Figure 4).

CRP, C-reactive protein; LDL/VLDL, low-density lipoprotein/very low-density lipoprotein; and TGFβ receptor II, Transforming growth factor beta receptor II.

## 4. Discussion

Endothelial cell dysfunction has been reported to persist in patients with sequelae of KD vasculitis [10,22,23,24]. However, there is no evidence of a direct relationship between KD vasculitis and early progression to atherosclerosis. In this study, we formed a KD-like vasculitis model of atherosclerosis with CAWS in Apo E-/- mice. Through this animal model of atherosclerosis, we confirmed that CAWS vasculitis promotes atherosclerosis. Subsequently, we showed in animal models that statins have anti-inflammatory and anti-atherosclerotic effects against atherosclerosis after CAWS vasculitis. Recently, Miyabe et al. reported that macrophage cell activation in response to CAWS deposits in the adventitia of the aortic root caused a large and persistent inflammatory cell infiltration around the coronary arteries [25]. In this study, a long-term infiltration of inflammatory cells, including macrophage cells, was also observed at the aortic root and in the abdominal aorta. TGFβ receptor II was strongly expressed in the vessel wall of the aortic root, supporting the presence of CAWS-induced vasculitis [21]. Statins significantly suppressed the infiltration of inflammatory cells, including macrophage cells, TGFβ receptor II expression, and atherosclerosis in the aorta. Atherosclerosis is a chronic inflammatory disease of the arterial wall, and macrophages have an important role as the starting point of plaque formation in patients with atherosclerosis [26,27].

Macrophages are broadly classified into the M1 phenotype, which produces inflammatory cytokines and has tissue-destructive properties, and the M2 phenotype, which produces anti-inflammatory cytokines and promotes angiogenesis and tissue repair [28] and whose phenotype and function depend on the microenvironmental stimuli [29]. We have previously reported that the phenotype of macrophages in coronary aneurysms of human KD patients differs from that of normal atherosclerotic lesions, with a predominance of the M1 phenotype. We have also reported that statins have an anti-inflammatory effect on the coronary atrial wall of patients treated with statins for more than 3 years [30]. In this study, most macrophages infiltrating the aortic root were the M1 phenotype. Statins might suppress the tissue infiltration of M1 macrophages or inhibit their differentiation into the M1 phenotype. Zhang et al. reported that statins induced monocyte differentiation into M2 macrophages [31]. This analysis showed that the M2 macrophage cell invasion area ratio was lower, and although statin administration increased the count of M2 macrophage cells, the difference between groups was insignificant. However, statins significantly suppressed the overall macrophage cell area, and when the M2/M1 macrophage cell area ratio was calculated, the M2/M1 ratio was significantly increased by statin administration. After 10 weeks of CAWS administration, the effect of statins on increasing M2 macrophages was also observed in the late-statin group, even though it was inferior to the group that had received statins from the start. The results suggested that statins might suppress inflammation and promote tissue repair by increasing M2 macrophages relatively.

In contrast to the 10 to 150 mg/dL serum LDL/VLDL cholesterol levels of non-Apo E-/- mice fed a high-fat diet [32,33,34], the serum LDL/VLDL levels of Apo E-/- mice fed a high-fat diet are known to be significantly higher than those of normal mice [27,33,34,35,36,37]. Interestingly, although the lipid-lowering effect of statins was confirmed in this analysis, the degree of reduction was mild. There were no animals whose LDL/VLDL cholesterol levels were normalized with statins. Statins have not only LDL-lowering but also pleiotropic effects, including anti-inflammatory effects, improvements in endothelial cell function, effects on macrophage fractions, and antioxidant effects [38]. It was considered that the therapeutic effect of statins observed in this analysis was more likely due to pleiotropic effects rather than lipid-lowering effects. The late-statin group in this analysis was designed to determine whether statins have a therapeutic effect in adult patients with KD who have started statins in young adulthood. Statins are not approved for infants, and it is difficult for patients with KD to take statins continuously from infancy. Remarkably, the late-statin group showed almost the same anti-inflammatory and anti-atherosclerotic effects as the group that was given statins after 2 weeks of CAWS administration (CAWS+statin group) (Figure 2a and Figure 3a). These results suggest that statins have a sufficient therapeutic effect, even if statins were started as patients with KD were adults.

In this study, it was confirmed that statins have an inhibitory effect on vascular inflammation and atherosclerosis in CAWS vasculitis-infected mice. However, this is an animal study using mice only. Moreover, CAWS vasculitis is not exactly the same as KD vasculitis, although CAWS vasculitis is considered an experimental model that has been generalized as KD-like vasculitis. Newly designed statin clinical trials should be conducted to establish the efficacy of statin against KD. Second, we have not been shown that CAWS vasculitis and statins affect vascular endothelial cell function. We have previously reported that KD vasculitis may impair endothelial cell function [10], and several centers have reported that statins may improve endothelial cell function and suppress vasculitis [12,13,39,40]. It is considered important to research the relationship between inflammatory cell invasion, including macrophages, in the remote stage and the statin-induced improvement of vascular endothelial cell function to elucidate the usefulness of statins in patients with KD.

## 5. Conclusions

It was found that the administration of CAWS to an atherosclerosis model mice induced KD-like vasculitis and that vasculitis promoted atherosclerosis in the remote stage of KD. Moreover, statin therapy inhibited the development of KD-like vasculitis, which induces atherosclerosis, and had an anti-inflammatory effect on the vasculitis itself. The results suggest that statin therapy is effective for patients with KD accompanied by cardiovascular sequelae. The indication for statin therapy in such patients may be extended in the future.

## Figures and Tables

**Figure 1 biomedicines-10-01794-f001:**
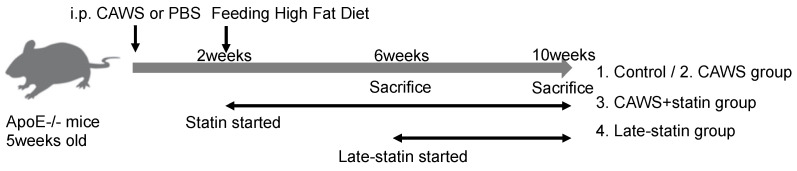
Experimental procedure. CAWS (4 mg for 5 consecutive days; control group received PBS instead) was injected intraperitoneally into 5-week-old Apo E-/- mice to induce KD-like vasculitis (the control group received phosphate-buffered saline instead). After 2 weeks of CAWS administration, the diet of all mice was changed to a high-fat diet to promote atherosclerosis. Mice were divided into four groups: 1. control group, 2. CAWS group, 3. CAWS+statin group, and 4. late-statin group. Statins (10 mg/kg/day) were dissolved in 0.5 mL of 0.5% Methylcellulose solution and then administered orally with a sonde. The control group was administered methylcellulose solution without statin. The CAWS and CAWS+statin groups received statins daily from 2 weeks after CAWS administration to the end of the experiment. The Late-statin group received statins daily from 6 weeks after CAWS administration to the end of the experiment. After 6 and 10 weeks of CAWS administration, mice were sacrificed, and the samples were collected.

**Figure 2 biomedicines-10-01794-f002:**
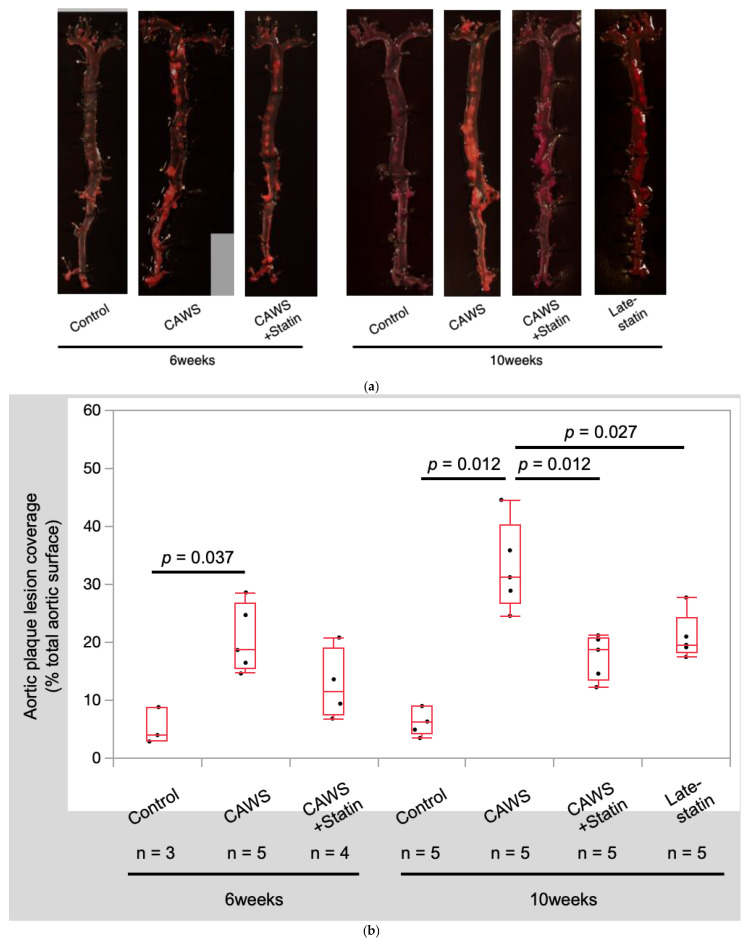
Histological findings of aortic plaque lesion coverage. The paraformaldehyde-fixed aorta was stained with Oil Red O. Quantification of aortic plaque lesion coverage area ratios on the entire aortic surface was performed in Apo E-/- mice. The plaque area was significantly higher in the CAWS mice than in statin-treated mice. Interestingly, the late-statin group showed a similar reduction in the aortic plaque area as the statin group. Total aortic plaque coverage ratios were calculated and compared between groups. Area ratios were expressed as percentages. CAWS significantly promoted atherosclerosis at both 6 and 10 weeks after administration. In both the late-statin and CAWS+statin groups at 10 weeks, statins significantly inhibited atherosclerosis. The inhibitory effects were comparable between both groups. (**a**) Quantification of aortic plaque lesion coverage area ratios on the entire aortic surface. (**b**) Total aortic plaque coverage ratios.

**Figure 3 biomedicines-10-01794-f003:**
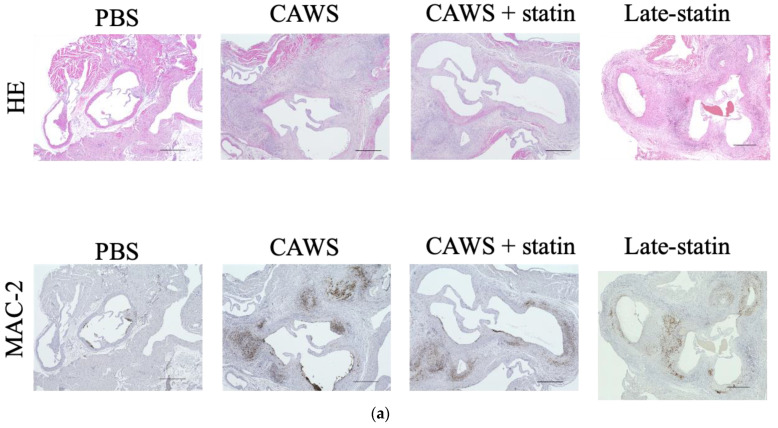
Histological findings at the aortic root. (**a**) Histological findings at the aortic root after 10 weeks of CAWS administration. Histological evaluation of inflammatory or macrophage cell invasion into the sinus area of the aortic root of Apo E-/- mice. Statin treatment suppressed the infiltration of inflammatory cells and macrophage cells at the aortic root; the anti-inflammatory effect of statins was also observed in the late-statin group. (**b**) Inflammatory cell invasion area and macrophage cell invasion area ratios. Inflammatory cell and macrophage cell invasion area ratios were calculated and compared between groups. Area ratios were expressed as percentages. CAWS promoted inflammatory and macrophage cell infiltration at both 6 and 10 weeks after CAWS administration. Statins also significantly inhibited the infiltration of inflammatory cells and macrophage cells at both 6 and 10 weeks after CAWS administration. In the late statin group, statins suppressed the inflammatory cell invasion area ratio. Means and SD are shown.Upper, H&E staining; lower, anti-MAC-2 antibody staining. (Bar = 500 μm).

**Figure 4 biomedicines-10-01794-f004:**
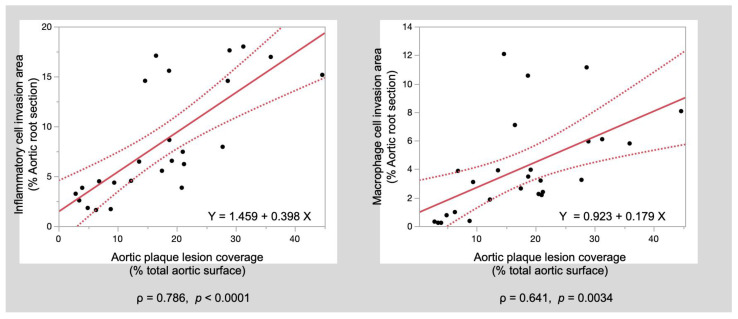
Correlation coefficient scatter diagrams. The aortic plaque lesion coverage ratio was significantly associated with the inflammatory cell invasion area ratio and macrophage cell invasion area ratio at the aortic root.

**Table 1 biomedicines-10-01794-t001:** Summary of the data at 6 and 10 weeks after CAWS administration.

**6 weeks after CAWS administration**														
	Control			CAWS			CAWS+statin							
Body weight (g)	28.4	±	1.2	28.9	±	2.4	27.1	±	1.4 *	**				
	(*n* = 7)			(*n* = 9)			(*n* = 8)							
Spleen weight (g)	0.14	±	0.02	0.23	±	0.02 *	0.22	±	0.04	**				
	(*n* = 7)			(*n* = 9)			(*n* = 8)							
Spleen/body weight ratio (%)	0.5	±	0.1	0.8	±	0.1 *	0.8	±	0.1 *					
	(*n* = 7)			(*n* = 9)			(*n* = 8)							
LDL/VLDL cholesterol (mg/dL)	472.5	±	45.0	543.0	±	6.8 *	528.3	±	11.6	**				
	(*n* = 4)			(*n* = 4)			(*n* = 4)							
hs-CRP (pg/mL)	102.7	±	20.9	307.4	±	121.5 *	248.2	±	40.1 *					
	(*n* = 4)			(*n* = 4)			(*n* = 4)							
Aortic plaque lesion coverage (%)	5.2	±	3.2	20.6	±	5.9 *	12.7	±	6.1					
	(*n* = 3)			(*n* = 5)			(*n* = 4)							
Aortic root														
Inflammatory cell invasion area (%)	2.7	±	1.0	15.4	±	1.2 *	4.8	±	1.2 *	**				
	(*n* = 4)			(*n* = 4)			(*n* = 4)							
Aortic root														
Macrophage cell invasion area (%)	0.3	±	0.1	10.2	±	2.2 *	3.5	±	0.4 *	**				
	(*n* = 4)			(*n* = 4)			(*n* = 4)							
Macrophage M1 cell area (%)	0.6	±	0.3	11.4	±	2.5 *	1.9	±	0.4 *	**				
	(*n* = 4)			(*n* = 4)			(*n* = 4)							
Macrophage M2 cell area (%)	0.2	±	0.2	1.1	±	0.5 *	2.1	±	0.3 *	**				
	(*n* = 4)			(*n* = 4)			(*n* = 4)							
TGFβ receptor II area (%)	1.1	±	0.7	25.7	±	3.9 *	5.7	±	1.4 *	**				
	(*n* = 4)			(*n* = 4)			(*n* = 4)							
Abdominal aorta														
Inflammatory cell invasion area (%)	3.3	±	1.2	9.5	±	0.9 *	4.2	±	0.4	**				
	(*n* = 4)			(*n* = 4)			(*n* = 4)							
Abdominal aorta														
Macrophage cell invasion area (%)	0.2	±	0.1	2.0	±	1.6 *	1.1	±	0.2 *					
	(*n* = 4)			(*n* = 4)			(*n* = 4)							
**10 weeks after CAWS administration**														
	Control			CAWS			CAWS+statin				Late-statin			
Body weight (g)	33.1	±	4.2	28.0	±	2.3 *	26.2	±	1.6*		26.9	±	1.5 *	
	(*n* = 8)			(*n* = 9)			(*n* = 9)				(*n* = 9)			
Spleen weight (g)	0.14	±	0.02	0.27	±	0.04 *	0.25	±	0.04 *		0.24	±	0.06 *	**
	(*n* = 8)			(*n* = 9)			(*n* = 9)				(*n* = 9)			
Spleen/body weight ratio (%)	0.4	±	0.1	1.0	±	0.1 *	0.9	±	0.1 *		0.9	±	0.2 *	**
	(*n* = 8)			(*n* = 9)			(*n* = 9)				(*n* = 9)			
LDL/VLDL cholesterol (mg/dL)	513.3	±	49.8	532.9	±	13.9	532.0	±	8.6 **		446.5	±	52.7 **	
	(*n* = 4)			(*n* = 4)			(*n* = 4)				(*n* = 4)			
hs-CRP (pg/mL)	79.4	±	17.8	396.3	±	197.5 *	314.6	±	108.9 *		141.0	±	86.0 **	
	(*n* = 4)			(*n* = 4)			(*n* = 4)				(*n* = 4)			
Aortic plaque lesion coverage (%)	6.5	±	2.4	33.0	±	7.6 *	17.5	±	3.9 *	**	21.0	±	4.0 *	**
	(*n* = 5)			(*n* = 5)			(*n* = 5)				(*n* = 5)			
Aortic root														
Inflammatory cell invasion area (%)	2.0	±	0.5	16.9	±	1.3	6.0	±	2.0 **		6.9	±	1.1 **	
	(*n* = 3)			(*n* = 4)			(*n* = 4)				(*n* = 4)			
Aortic root														
Macrophage cell invasion area (%)	0.7	±	0.4	6.5	±	1.1	2.5	±	0.7 *	**	3.0	±	0.8	**
	(*n* = 3)			(*n* = 4)			(*n* = 4)				(*n* = 4)			
Macrophage M1 cell area (%)	1.0	±	0.3	6.3	±	1.3 *	1.9	±	0.3 *	**	2.1	±	0.7 *	**
	(*n* = 3)			(*n* = 4)			(*n* = 4)				(*n* = 4)			
Macrophage M2 cell area (%)	0.2	±	0.1	1.0	±	0.4	1.8	±	0.3 *		0.9	±	0.2	
	(*n* = 3)			(*n* = 4)			(*n* = 4)				(*n* = 4)			
TGFβ receptor II area (%)	0.4	±	0.3	17.5	±	2.5 *	6.2	±	1.8	**	7.6	±	1.4 *	**
	(*n* = 3)			(*n* = 4)			(*n* = 4)				(*n* = 4)			
Abdominal aorta														
Inflammatory cell invasion area (%)	3.3	±	0.7	11.1	±	1.9 *	4.6	±	0.9	**	5.8	±	0.6	
	(*n* = 3)			(*n* = 4)			(*n* = 4)				(*n* = 4)			
Abdominal aorta														
Macrophage cell invasion area (%)	0.2	±	0.2	6.3	±	3.4 *	2.1	±	1.9 *		2.0	±	2.3	
	(*n* = 3)			(*n* = 4)			(*n* = 4)				(*n* = 4)			

* *p* < 0.05 compaird to Control. ** *p* < 0.05 compaird to CAWS.

## Data Availability

Not applicable.

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
