# Peer review of "Kawasaki Disease-like Vasculitis Facilitates Atherosclerosis, and Statin Shows a Significant Antiatherosclerosis and Anti-Inflammatory Effect in a Kawasaki Disease Model Mouse"

_biomedicines, 2022, doi:10.3390/biomedicines10081794_

Round 1

Reviewer 1 Report

Comments to the Author

Title: Kawasaki Disease-like Vasculitis Facilitates Atherosclerosis, and Statin Shows a Significant Antiatherosclerosis and Anti-Inflammatory Effect in a Kawasaki Disease Model Mouse

Authors: Yusuke Motoji, Ryuji Fukazawa, Ryosuke Matsui, Noriko Nagi-Miura, Yasuo Miyagi, Yasuhiko Itoh, Yosuke Ishii

à The present work aims to examine the relationship between KD, 13 atherosclerosis, and the long-term effects of HMG-CoA inhibitors (statins). The authors shown that the long-term administration of statin (via oral) highly suppressed both atherosclerosis and inflammatory cell infiltration. They conclude that statin treatment may be used for the secondary prevention of cardiovascular events in the chronic phase of KD.

à I believe this is an interesting work and is adequate for publication in Biomedicines, however some major changes are required. 

I would consider the publication of the present manuscript after carefully addressing the comments hereafter and modifying its structure. 

General: The manuscript grammar and presentation require to be double checked since can’t be easily followed by the reader. The section length is adequate in most of the sections but, the presentation of the results should be improved.  The reference list is a bit short.

In the following: MA=Major comment, MI = Minor comment, OP = Optional Comment

(MI) P1, Abstract, L12-26: I encourage the authors to introduce connectors in the abstract. When reading this paragraph, it feels like reading a bullet list instead of an abstract. Please, re-write.

(MI) P1, Introduction, L33: In which year or which period of time?

(MA) P1, Introduction, L33-35: What about the others? You are just classifying 165,000 patients when a total of 360,000 is mentioned. Please, clarify.

(MI) P1, Introduction, L38: Add reference.

(MI) P1, Introduction, L41: Add reference.

(MI) P2, Introduction, L50: What does it mean “minor studies”?

(MI) P2, Introduction, L61: Please, mention where the study was performed.

(MI) P2, Material and Methods, L50: What does it mean “minor studies”?

(MI) P2, Material and Methods, L78: Add reference.

(MI) P2, Material and Methods, L80-81: For how long were you feeding the mice with the high-fat diet?

(OP) P2, Material and Methods, L83-90: You may consider adding a sketch with the preparation steps. This type of presenting the process is usually helpful for the reader.

(MI) P3, Material and Methods, L104-105: Was the same volume of methylcellulose than of statin injected?

(MI) P3, Material and Methods, L104-114: Did you use the same number of mice for every group?

(OP) P3, Material and Methods, L129: The full information was already provided in the previous section. There is no need to include all the information again.

(MI) P3, Material and Methods, L133: Please, explain why investigating the TGFβ receptor expression is of interest.

(OP) P3, Material and Methods, L134-139: It would be better providing this information in a table.

(MI) P4, Material and Methods, L154: Add reference to Kruskal-Wallis.

(MI) P4, Results, L166-174: Are your measurements precise enough to provide two significative decimal digits? Maybe you should round your values to just one decimal.

(MI) P5-P6, Results: My previous comments apply to all reported results.

(MI) P7, Results, Figure 1: Figure captions should be self-explanatory. Please make the explanations more complete.

(MI) P7, Results, Figure 1: The acronym wks has not been defined nor used before. Please introduce it the first time you use the word weeks and use it through the manuscript. Otherwise modify the acronym in the Figure.

(MI) P9, Results, Figures: I have notice that you are explaining the Figures after the caption. The explanation should be embedded in the caption. 

(MI) P10, Results, Figures 3, L 380: What are the stars referring to?

(MI) P11, Results, Figures 4: It is tough to read this Figure labels. Please enlarge the text.

(MA) P12, Results, Table I: All values presented in the table should report the same number of significative digits.

(MA) P13, Results, Table II: The previous comment also applies here.

(MI) P14, Discussion, L423: In the previous sections of the manuscript you were not including the hyphen for the TGF-β receptor. Please, homogenize styles.

(MI) P14, Discussion, L477-480: Are you planning to perform these studies?

(MI) P15, Discussion, L480: Please, discuss the impact of your conclusions.

(MI) P15, Discussion, L488: How do you plan to extend the indication for statin therapy in such patients?

 (MA) P16-17, References: Check that all references include all relevant information such as Authors, Volume, Page number…

Also, verify that the reference style follows the journal guideline. 

Reviewer 2 Report

Thank you for sharing your data. I have some comments and questions.

1. Kawasaki disease is a common disease in childhood, especially in early childhood. On the other hand, dyslipidemia is rare in early childhood. In the introduction of this study, it is necessary to show the premise that patients with Kawasaki disease have dyslipidemia.

2. How old is the mice (5, 7, 13, or 17 weeks) in this study in human years?

3. Please show the representative data of how the authors measured the plaque coverage area, cell invasion area, M1 cell, M2 cell, TGFbR2 antibody, or MAC-2 antibody with supplementary figures.

4. Please clarify the adverse events of statins, especially in the musculoskeletal system, in this study. Also discuss the side effects of statins. 

5. How could the results of this study change the pediatrician's practice for patients with Kawasaki disease?

6. Which of the two is Y.I. in the author's contribution? 

Round 2

Reviewer 1 Report

Dear authors,

Thanks for considering my review.

Reviewer 2 Report

None.